# 11β-HSD as a New Target in Pharmacotherapy of Metabolic Diseases

**DOI:** 10.3390/ijms23168984

**Published:** 2022-08-11

**Authors:** Daria Kupczyk, Rafał Bilski, Mariusz Kozakiewicz, Renata Studzińska, Kornelia Kędziora-Kornatowska, Tomasz Kosmalski, Agnieszka Pedrycz-Wieczorska, Mariola Głowacka

**Affiliations:** 1Department of Medical Biology and Biochemistry, Collegium Medicum in Bydgoszcz, Nicolaus Copernicus University in Toruń, Karłowicza 24, 85-092 Bydgoszcz, Poland; 2Department of Geriatrics, Nicolaus Copernicus University in Toruń, L. Rydygier Collegium Medicum in Bydgoszcz, Dębowa 3, 85-626 Bydgoszcz, Poland; 3Department of Organic Chemistry, Collegium Medicum in Bydgoszcz, Nicolaus Copernicus University in Toruń, Jurasza 2, 85-089 Bydgoszcz, Poland; 4Department of Histology and Embriology, Medical University in Lublin, Radziwiłłowska 11, 20-080 Lublin, Poland; 5Faculty of Health Sciences, Mazovian State University in Płock, Plac Dąbrowskiego 2, 09-402 Płock, Poland

**Keywords:** 11β-HSD enzyme, 11β-HSD1 inhibitors, metabolic syndrome, glucocorticoids, diabetes mellitus, obesity

## Abstract

Glucocorticoids (GCs), which are secreted by the adrenal cortex, are important regulators in the metabolism of carbohydrates, lipids, and proteins. For the proper functioning of the body, strict control of their release is necessary, as increased GCs levels may contribute to the development of obesity, type 2 diabetes mellitus, hypertension, cardiovascular diseases, and other pathological conditions contributing to the development of metabolic syndrome. 11β-hydroxysteroid dehydrogenase type I (11β-HSD1) locally controls the availability of the active glucocorticoid, namely cortisol and corticosterone, for the glucocorticoid receptor. Therefore, the participation of 11β-HSD1 in the development of metabolic diseases makes both this enzyme and its inhibitors attractive targets in the pharmacotherapy of the above-mentioned diseases.

## 1. Regulation of the Local Action of Glucocorticoids

Glucocorticoids (GCs) are steroid hormones secreted by the cortex of the adrenal glands. They play an important regulatory role in the metabolism of carbohydrates, lipids, and proteins. In addition, they modulate the immune response, influence the regulation of blood pressure, and maintain the water and electrolyte balance. They also affect changes in mood, behavior, neuroendocrine functions, body temperature, and pain perception. GCs are made from cholesterol by the adrenal cortex [1,2,3]. Their secretion is controlled by the hypothalamic–pituitary–adrenal (HPA) axis. Moreover, most of the physiological and pharmacological actions of glucocorticosteroids are mediated by a specific receptor, which is involved in the transmission of information contained in the molecule and acts as a transcription factor. Its site of action is the cytoplasm, so it indicates the presence of the cytoplasmic mechanism of action of glucocorticosteroids [3].

The hypothalamus is responsible for the involvement of circadian, physical, or emotional factors, as it activates the pathway that leads to GC synthesis. Namely, the secretion of corticoliberin (CRH) and vasopressin (AVP) causes the stimulation of the production of the proopiomelanocortin peptide, from which the adrenocorticotropic hormone (ACTH) is released into the blood after the transition from the hypothalamus to the anterior pituitary gland [4]. It is the ACTH that, by affecting the adrenal cortex, leads to the synthesis of cortisol/corticosterone. The factor that limits the synthesis of steroid hormones is the rate at which cholesterol passes through the mitochondrial membrane. Thus, on the one hand, the relationship between the secretion of GCs and adrenocorticotropic hormone is emphasized, and on the other hand, it indicates that, inter alia, adipokines, cytokines, and growth factors may influence GC secretion independently of ACTH [5,6].

## 2. Isoforms Regulating the Local Action of Glucocorticoids

Cortisol is the dominant human GC, whereas corticosterone predominates among rodents. Both cortisol and corticosterone are among active GCs. The presence of a hydroxyl group in the C11 position of the steroid molecule influences the biological activity of GC [7]. As a result of the oxidation of this group, the compound is transformed into an inactive form, which includes cortisone and 11-dehydrocorticosterone. The initial mentions of 11β-hydroxysteroid dehydrogenase type 1 (11β-HSD1) are related to the date of discovery of cortisone. In addition, further findings are related to the interconversion of cortisol in humans and corticosterone in rodents, respectively, which is closely related to the activity of 11β-HSD. In the late 1990s, two isoforms (11β-HSD1, 11β-HSD2) of this enzyme were described. The scheme of their operation is presented in Figure 1 [8].

11β-HSD1 is a product of the HSD11B1 gene, which is located on chromosome 1 in humans and rodents. 11β-HSD1 catalyzes the interconversion of cortisone to cortisol. In turn, 11β-HSD2 is encoded by a gene that is located on chromosome 16 in humans and on chromosome 8 in rodents, respectively. Both types of 11β-hydroxysteroid dehydrogenase are microsomal enzymes that are associated with the cell membrane of the endoplasmic reticulum. 11β-HSD1 is expressed in liver, adipose tissue, brain, vessels, and gonads. Its role is to increase the concentration of the active form of GCs in the tissue, leading to the activation of the GC receptor. In turn, 11β-HSD2 occurs in tissues associated with the action of mineralocorticoids, i.e., kidney, large intestine, placenta, and salivary glands. The enzyme has not been found in adipose tissue. Its action leads to the inactivation of GCs and prevents the activation of the mineralocorticoid receptor. Thus, 11β-HSD2 leads to the rapid inactivation of cortisol. It is these enzymes described above that play a significant role in the peripheral mechanism of cortisol generation [10,11].

Both 11β-HSD1 and 11β-HSD2 contain an N-terminal sequence in their structure that allows them to be located in the endoplasmic reticulum membrane. The catalytic part of 11β-HSD1 is directed precisely towards the lumen of the endoplasmic reticulum, while in the case of 11β-HSD2, it is directed towards the cytoplasm (Figure 2).

The activity of 11β-HSD1 requires the participation of hexose-6-phosphate dehydrogenase. Both enzymes are found in the membrane of the endoplasmic reticulum. The amount of NADP^+^ and NADPH available determines the direction of the reaction catalyzed by 11β-HSD1. In the purified state, this enzyme shows a two-way effect, while in the presence of reducing NADPH, which is formed precisely with the participation of hexose-6-phosphate dehydrogenase, it switches to reductase activity and generates the secretion of active GCs. In addition, it is this isoform that exhibits dimer activity and is characterized by cooperative kinetics both for cortisone and 11-dehydrocorticosterone, which explains its adaptation to nanomolar and microsomal concentrations.

## 3. The Role of Glucocorticoids in Metabolic Syndrome

GCs play a significant role in maintaining homeostasis of the body; hence, changes in the activity of both isoforms of 11β-hydroxysteroid dehydrogenase may have an adverse effect on the course of metabolic processes and thus contribute to the development of, among others, obesity, type 2 diabetes, cardiovascular diseases, or arterial hypertension [12,13,14].

Considerations about the influence of GCs on the cell begin with emphasizing their influence on the cell membrane. It turns out that these compounds in high concentrations may contribute to the change of the physicochemical properties of biological membranes, namely the cell and mitochondrial membranes. GCs integrate into these membranes, thereby changing the functions performed by membrane proteins. This, in turn, has a significant impact on the lipid peroxidation process and leads to a change in membrane permeability. The interaction of GCs with the cell membrane of the immune system cells leads to a decrease in the flow of Na^+^ and Ca^2+^ ions through the membrane and an increase in immunosuppression processes, resulting in a reduction of inflammatory processes. Moreover, it was also observed that the participation of GC increases the release of protons from the mitochondria and thus leads to a decrease in the production of ATP [15]. Changes in ATP production may, in turn, contribute to the development of cell dysfunctions. Thus, strict control of the release of GCs and regulation of their cellular activity play a key role in the proper regulation of metabolism in response to changing environmental conditions, which is of particular importance in the states of both deficiency and excess of GC. Recent studies have linked the excess of GCs with the development of metabolic disorders such as obesity, dyslipidemia, hyperglycemia, insulin resistance, or hypertension, i.e., components of the metabolic syndrome. It is emphasized that long-term and persistent elevated concentration of GC may contribute to the development of the above-mentioned disorders [16,17,18,19,20].

It turns out that the local concentration of GCs, which is regulated by 11β-HSD, is more important than the systemic increased level of GC [17].

Obesity is described as a chronic disease of complex etiology. It is based on genetic, environmental, and endocrine predispositions. Increase in the amount of adipose tissue in the body leads to pathology, and dysfunction of individual systems or organs develops. Obesity is called an epidemic and is one of the main causes of morbidity and mortality in developed countries. It is a factor in the development of, among others, type 2 diabetes, high blood pressure, and metabolic syndrome. The problem of obesity treatment and its effects in the body pose a challenge for health care, especially in the face of an aging population. In addition, age and comorbidities, including obesity, are the main risk factors and the severity of the disease caused by the SARS-CoV-2 virus. Incorrect acting of immune system during SARS-CoV-2 virus may lead to the release of inflammatory cytokines without any regulation mechanisms. The reasons for such a condition can be found both in the abnormal immune response of the body as well as in differences in the concentration of hormones in these people or in elevated cholesterol levels [21,22,23]. This is all the more important, as additionally, during the pandemic, an increase in obesity-related morbidity and a deterioration in the health of patients is observed. The therapeutic target in the elderly is even more difficult to achieve due to the deteriorating general health condition with age [24,25].

In human adipose tissue, glucocorticoids induce adipocyte differentiation leading to increased adiposity and insulin resistance. A classic example of the relationship between GC level and obesity is Cushing’s syndrome, in which the elevated level of cortisol leads to visceral obesity [26]. In humans, there are some studies suggesting that long-term elevations of endogenous circulating cortisol is associated to obesity. A study performed by Roberts et al. showed that increased salivary cortisol levels across a university study semester coincides with a greater increase in BMI. The participants of this study were 71 healthy women who were studying university-based nursing practitioner program. Over the course of the semester, during 12 weeks of classes and an exam session, the researchers measured, among others, weight, BMI, and salivary cortisol secretion. Most of the participants gained weight over the 12-week period, which affected the BMI results. Salivary cortisol secretion was also elevated in comparison to the results obtained before the 12-week trial [27].

As a result of the stress experienced by the body, such as malnutrition or hunger, glucocorticoids maintain the proper level of glucose in the blood to maintain homeostasis. The excess of glucocorticoids contributes to the development of the problem of insulin resistance, hyperglycemia, dyslipidemia, and obesity, i.e., components of the metabolic syndrome. Patients with this disease have an increased level of GC in adipose tissue, which is related to the high activity of the 11BHSD1 enzyme, which transforms cortisone into its active form [28,29]. In turn, excessive activation of 11BHSD2 in adipose tissue protected the mice tested against diet-induced obesity [30].

Adipose tissue is essential for the maintenance of homeostasis in the body. Although the exact mechanisms of the participation of GCs in the above-mentioned in vivo process are not known, it is known that they play a significant role in the differentiation of adipocytes in vitro. In addition, GCs affect the fat metabolism of adipose tissue by promoting lipolysis in white adipose tissue during fasting. This process leads to the production of glycerol and fatty acids, which in turn enhances gluconeogenesis in the liver [31,32]. However, in pathological conditions such as, for example, Cushing’s disease, fat metabolism in adipose tissue is much more complicated. People suffering from this disease develop central obesity while reducing subcutaneous white adipose tissue, which in turn translates into the formation of lipodystrophy [33]. Changes in intracellular GC metabolism in the pathogenesis of obesity emphasize the role of selective 11β-HSD1 inhibition as a new therapeutic target in the pharmacotherapy of this disease.

After 20–30 years of age, the mass of fat begins to increase, which lasts until about 60–70 years of age. After this period, both muscle mass and fat decrease. During the aging process, adipose tissue is redistributed, and the amount of visceral fat increases also in the muscles in relation to subcutaneous fat and the total mass of adipose tissue [34]. It is the increase in adipose tissue in skeletal muscles and in the liver that leads to the development of type 2 diabetes, insulin resistance, hypertension, and the development of cardiovascular diseases [35]. In addition, an aging body causes hormonal changes that may favor the formation of changes in adipose tissue [36]. Obesity in the elderly is also accompanied by many metabolic changes. The frequency of occurrence of the components of the metabolic syndrome, such as glucose intolerance, insulin resistance and hyperinsulinemia, dyslipidemia, hypertension, and ischemic heart disease, is increasing. The increase in adipose tissue mass is responsible for the appearance of these changes [37,38,39,40].

Insulin is a hormone produced by the β cells of the pancreas. It is released in small amounts into the blood after each meal, which helps to transport glucose to the cells, where it is used as an energy source. Insulin resistance is a condition in which the effect of insulin on target tissues is reduced despite normal or elevated levels of insulin in the blood serum. Since glucose is needed for cells to survive, the body begins to compensate for this by producing more insulin. This results in an excess of insulin in the blood and an over-stimulation of tissues that remain sensitive to the action of this hormone. The amount of glucose and insulin is disturbed in the body [41,42].

Insulin resistance may lead to abnormalities in fat metabolism in the body and thus to an increase in the number of triglycerides and LDL cholesterol in the blood. The most common diseases with insulin resistance include type 2 diabetes, arterial hypertension, and renal failure [43,44,45]. As a result of excess insulin in the blood and excessive adipose and muscle tissue stimulation, the balance between the amount of glucose and insulin is disturbed. On the other hand, when the production of insulin is insufficient, there is an increase in hyperglycemia and the development of type 2 diabetes, a disease leading to multi-organ damage [42]. Another group of disorders associated with the metabolic syndrome are hypercholesterolemia and atherogenic dyslipidemia, which includes the triad of lipid disorders, i.e., increased triglycerides, decreased HDL cholesterol, and the presence of the so-called small, dense particles of LDL cholesterol fraction [46,47,48,49,50]. These disorders are also a significant problem in elderly patients, as they may increase the risk of death due to cardiovascular diseases. An incredibly significant issue and a challenge for 21st century medicine is also the growing number of cases of type 2 diabetes. This disease causes premature mortality, mainly due to cardiovascular complications related to limb amputation, kidney failure, or retinopathy. A long-term increase in blood glucose levels can damage blood vessels in the kidneys, eye, brain, heart, and nerves. This disease is revealed in adulthood, and its incidence increases with age. It is not uncommon for a patient diagnosed with type 2 diabetes to also suffer from hypertension, lipid metabolism disorders, or symptoms of ischemic heart disease [51,52,53,54,55]. Thus, the multitude of the above-mentioned disease entities and disorders as well as their risk factors and effects mutually reinforce and add up, increasing the risk of dangerous complications. Complications due to the existing arterial hypertension are dangerous because they can lead to chronic kidney failure and, consequently, damage to this organ, heart failure, heart attack, and ischemic stroke. Conducting research aimed at elucidating the tissue-specific mechanism of GC action and the participation of 11β-HSD may be of significant importance for the pathogenesis of the above-mentioned metabolic diseases and allow for the achievement of new goals in pharmacotherapy [56]. This is especially important due to the fact that epidemiological data concerning the prevalence of the components of the metabolic syndrome are not optimistic due to the increasing frequency of their occurrence taking the form of an epidemic [57,58]. Therefore, the development of hypotheses that increased levels of GCs due to the action of 11β-HSD1 may be the cause of obesity, type 2 diabetes, hypertension, or cardiovascular diseases is not without significance [59,60].

## 4. Metabolic Effects of 11β-HSD1 Deficiency

11β-HSD1 deficiency has metabolic consequences. In order to fully understand the role of this enzyme in the pathogenesis of the metabolic syndrome, mouse models were constructed that lacked the 11β-HSD1 gene (knock out). These mice were resistant to hyperglycemia, which was induced by a high-fat diet and stress. Moreover, in these mice, decreased triglycerides concentration, increased concentration of HDL cholesterol, as well as decreased fibrinogen synthesis in the liver were observed, which was caused by decreased expression of the 11β-HSD1 gene in the liver. In the adipose tissue of the mice studied, a decreased concentration of corticosterone was observed, with its slightly elevated level in the blood, and the application of a high-fat diet resulted in the peripheral distribution of adipose tissue [58,61]. The role of factors involved in the tissue-specific regulation of 11β-HSD1 expression is also important. They include, among other GCs, insulin and leptin, which participate in the regulation of the transcription of this enzyme. Differences in the availability of the NADPH cofactor affect the post-translational modification of the directions of action of 11β-HSD1. This is due to mutations in the hexose-6-phosphate dehydrogenase gene, which controls the availability of NADPH in the endoplasmic reticulum and thus influences the direction of the reaction catalyzed by 11β-HSD1 [62,63]. Therefore, considering the reports on the beneficial effects of reducing the activity of 11β-HSD1, attempts are made to use inhibitors of this enzyme for therapeutic purposes in the treatment of, among others, type 2 diabetes. Reports in the literature on the beneficial effect induced by a decrease in the activity of 11β-HSD1 caused the search for the use of those enzymes inhibitors for therapeutic purposes, including in the treatment of type 2 diabetes. Namely, the first proposed blocker was arylsulfonamidothiazole, a substance that inhibits the action of 11β-HSD1 in a non-selective manner. Another non-specific inhibitor of this enzyme is carbenoxolone. It has been shown that it increases insulin sensitivity in the liver of healthy subjects and patients with type 2 diabetes without increasing the peripheral glucose consumption [64]. In the studies by Sandeep et al. on the use of carbenoxolone in obesity, no changes between the study group and the placebo were observed in glucose concentration, its consumption, production rate, and insulin concentration [65]. In the study of Dhanesh et al., it was found that carbenoxolone lowers the level of adipose tissue, lipid profile, and glucose tolerance in obese mice [66]. In the study conducted by Heise et al., the researchers stated that the inhibitors of 11β-HSD1 may have beneficial effect in the treatment of diabetes mellitus and metabolic syndrome. The study was a placebo-controlled, randomized, double-blinded trial over 4 weeks. The participants were 110 male and female subjects aged 37–65 years. They were given two inhibitors of 11β-HSD1, RO-151, and RO-838 with metformin. Both inhibitors were well-tolerated by the study group, and any adverse effects occurred both in placebo-controlled and inhibitor dosed group. The results of the study showed improved parameters such as body weight, HbA1c, and insulin sensitivity. The comparison between those two inhibitors showed that RO-151 had higher inhibition index, but the dosage of RO-151 increased ACTH plasma levels. However, these findings suggest prolonging the studies for more than 4 weeks to show potential benefits and risks of studied compounds [67]. Feig et al. conducted research regarding 11β-HSD1 inhibitors in patients with diabetes mellitus and metabolic syndrome for 12 weeks. It was a randomized, double-blinded, placebo-controlled research conducted in the group of 154 patients. The participants were taking different doses of the 11β-HSD1 inhibitor MK-0916. The longer study showed positive changes in body weight and blood pressure and modest positive changes in glycated hemoglobin levels, with no statistically significant changes in fasting plasma glucose levels. The highest doses of MK-0916 showed increased levels of androstenedione and DHEA. The levels of androgens were normalized 3 weeks after the end of the study. The results of the study showed that MK-0916 was well-tolerated by the patients and without serious adverse effects [68]. In addition, Rosenstock et al. also conducted randomized, double-blinded, placebo-controlled research for 12 weeks. The study group consisted of 302 patients with type 2 diabetes mellitus on metformin monotherapy. The participants were given the 11β-HSD1 inhibitor INCB13739. The results of the study showed decreased total cholesterol level, LDL and triglycerides, HbA1c levels, fasting plasma glucose, body weight, and homeostasis model assessment–insulin resistance in comparison to placebo group. The study showed reversible increase in corticotrophin levels but no significant changes in androgen levels. The inhibitor administration showed no severe adverse effects [69]. The research for INCB13739 clinical evaluation of INCB13739 confirmed for the first time that tissue-specific inhibition of 11β-HSD1 in patients with type 2 diabetes mellitus was efficacious in controlling glucose levels and reducing cardiovascular risk factors [70]. Shah et al. studied MK-0916 and MK-0736 selective inhibitors of 11β-HSD1 in 249 obese patients aged 18–75 years with hypertension for 12 weeks. It was a randomized, double-blinded, placebo-controlled study. The results showed modest positive changes in patients’ body weight and LDL-C level. However, administration of MK-0736 resulted in decreased levels of HDL-C in comparison with placebo group. Both inhibitors were proved to be safely administrated without severe adverse effects although the study showed increased level of androgens in patients who were administered with inhibitors [71].

Hardy et al. conducted a phase II, randomized, double-blinded, placebo-controlled trial of a 12-week treatment in 31 obese women aged 18–55 years with idiopathic intracranial hypertension. The activity of the reversible inhibitor of 11BHSD1, AZD4017, and its influence on a number of parameters such as pro-inflammatory cytokines, adipokines, lipid profile, percentage of muscle and fat mass, and BMI. Elevated HDL cholesterol and decreased cholesterol/HDL ratio was achieved after 12 weeks of trial. However, both the level of triglycerides and the parameters of glucose homeostasis, including the levels of HbA1c, remained unchanged. The conducted studies also assessed the effect of the AZD4017 inhibitor on the condition of the liver and kidneys. There were no significant differences between the placebo group and the potential therapeutic group in terms of parameters such as bilirubin, ALT, and AST. On the other hand, GGT and ALP decreased but still remained within the reference values. There were also no changes in the pro-inflammatory parameters. Researchers also showed an increase in lean muscle mass after 12 weeks of administration of the inhibitor, while BMI and body fat mass remained unchanged. The androgen levels in the blood of the patients also increased slightly, while the level of cortisol decreased. The increase in androgen concentration correlated positively with the increase in the percentage of muscle mass. The results of the discussed studies indicate a positive and safe effect of the analyzed compound, which is another step on the way to introducing an effective drug [72]. In contrast, studies by Ajjan et al. about the AZD4017 inhibitor in a phase II, randomized, double-blinded, placebo-controlled trial were conducted in adults with type 2 diabetes mellitus without foot ulcers for 35 days. The researchers focused on the problem of wound healing in diabetic patients and the influence of AZD4017 on this process. Wound healing was monitored after 2 and 7 days of drug administration. The results showed a decrease by 34% in wound diameter at day 2, with wounds being 48% (12–85%) smaller after repeat wounding at day 30. There were minimal adverse effects in comparison with placebo group. The researchers concluded that the results are so promising that the study should be extended to the group of patients with foot ulcers [73].

In turn, Bianzano et al. analyzed the effects of another 11BHSD1 inhibitor, BI 187004, in 72 healthy men aged 18–55 years with obesity or overweight. The participants were divided into subgroups with different dosage of the inhibitor. The research was conducted as a randomized, double-blinded, placebo-controlled trial and focused on safety, tolerability, and pharmacokinetic and pharmacodynamic profiles of single rising doses of the above-mentioned inhibitor. The results shows that although nine participants had drug-related adverse effects, it was well-tolerated and safe in all tested dose groups. Median inhibition of 11beta-HSD1 in subcutaneous adipose tissue biopsies following single dosing ranged from 86.8% (10 mg) to 99.5% (360 mg) after 10 h and from 59.4% (10 mg) to 98.6% (360 mg) after 24 h [74]. Likewise, Bellaire et al. conducted phase I clinical, randomized, double-blinded, placebo-controlled trials assessing the safety of the 11BHSD1 inhibitor ASP3662. The study compared the results between groups of healthy young people aged 18–55 years outside Japan, healthy young people aged 18–55 years from Japan, and older people aged >65 years outside Japan. The study focused on two dosage forms: 48 participants in single ascending dose (SAD) and 62 participants in multiple ascending dose (MAD). The optional therapeutic drug was administered for 14 days, during which no adverse effects were observed, which allowed the investigators to confirm the safety of the use of the ASP3662 inhibitor [75].

## 5. 11β-HSD1 Inhibitors in the Therapy of Components of the Metabolic Syndrome

11β-HSD1 inhibitors partially inhibit the conversion of cortisone to cortisol, thereby affecting cortisol levels. Moreover, it is believed that they selectively lower the concentration of cortisol within the tissue without changing its plasma level during the stress response [76]. Numerous studies have also been conducted in which an increase in the expression of 11β-HSD1 was observed in obese patients compared to lean controls [77,78,79]. Higher levels of 11β-HSD1 mRNA have been observed in subcutaneous adipose tissue and visceral adipose tissue [80,81,82]. No significant differences in serum 11β-HSD1 expression were observed between the group of obese men with diabetes and the group of non-obese men [83]. Alberti et al. described the expression of 11β-HSD1 in people with metabolic syndrome and obesity. Obese subjects with the metabolic syndrome were characterized by a higher level of 11β-HSD1 expression in adipose tissue and a higher 11β-HSD1 mRNA activity in subcutaneous adipose tissue compared to obese subjects without metabolic syndrome [84]. On the other hand, studies by Baudrand et al. showed no differences in the expression of 11β-HSD1 in obese people with metabolic syndrome compared to obese people without metabolic syndrome [85]. Devang et al. conducted a genome study in diabetes and metabolic syndrome and compared the results with a control group devoid of these diseases. They were studying two polymorphisms of 11β-HSD1 (rs12086634 and rs846910). The results showed that the 11β-HSD1 rs12086634 is associated with diabetes mellitus and metabolic syndrome, while 11β-HSD1 rs846910 was only related to diabetes mellitus [86]. The study did not include people with concomitant metabolic syndrome and diabetes.

The results of studies that assess the expression or inhibitors of 11β-HSD1 indicate areas that require further analysis and distinguish those where it would be possible to achieve a potential therapeutic benefit. However, it is still not well-understood. Nevertheless, it has been suggested that in obesity, there is an increase in 11β-HSD1 expression or activity in adipose tissue. In turn, there is lower or no change in 11β-HSD1 expression in the liver [87,88]. Similarly, studies using inhibitors of this enzyme have not allowed to obtain conclusive results. Treatment with carbenoxolone also had an effect on glucose production but had no effect on obesity or parameters related to diabetes. From the medical point of view, studies in patients with concomitant metabolic syndrome, diabetes, and obesity seem to be important, but it should be borne in mind that the overlapping of these disease entities may cause difficulties in accurately determining the effect of 11β-HSD1 inhibitors. The researchers are focused on finding the new selective inhibitors for 11β-HSD1 although most of the obtained compounds have not been studied in humans yet. Liu et al., in their research, aimed at developing a pharmacological 11β-HSD1 receptor that would show specific activity in relation to adipose tissue proteins. Researchers studied mice and watched whether a selective enzyme inhibitor (BVT.2733) would reduce weight gain and improve glucose tolerance (Figure 3). The peptide-coupled inhibitor was administered to mice by subcutaneous injection over a two-week period. As a result, a decrease in body weight gain and adipocyte size and improvement in glucose tolerance were observed as compared to the control group [89]. Okazaki et al. synthesized HIS-388 (*N*-[(1R, 2S, 3S, 5S, 7S)-5-hydroxyadamantan-2-yl]-3-(pyridin-2-yl)isoxazol-4-carboxamide), which is an inhibitor of 11β-HSD1. In studies with mouse models of hypercortisolemia and hyperinsulinemia, a single administration of HIS-388 resulted in a strong and prolonged reduction in cortisol production and a decrease in plasma insulin levels. Researchers observed that the obtained effects were stronger than with other 11β-HSD1 inhibitors, namely carbenoxolone and compound 544 3-[(1S, 3S)-adamantan-1-yl]-6,7,8,9-tetrahydro 5*H*-[1,2,4] triazolo[4,3-*a*]azepine, indicating that HIS-388 strongly inhibits the enzyme activity in vivo. In diet-induced obese mice, this inhibitor significantly reduced fasting blood glucose and plasma insulin and improved insulin resistance. In addition, it led to weight loss and inhibited the elevated blood glucose levels during the pyruvate tolerance test. HIS-388 also significantly decreased postprandial blood glucose and plasma insulin levels in mice that were not genetically diabetic and had disease caused by a high-fat diet and a low dose of streptozotocin. The obtained results indicate that this inhibitor has a strong effect against type 2 diabetes. Moreover, the alleviation of diabetes symptoms by this inhibitor is related to the activity directed against obesity and the improvement of insulin resistance [90]. Hermanowski et al. administered a potent and selective 11β-HSD1 inhibitor, which was compound 544, leading to a reduction in body weight, insulin, glucose, triglycerides, and cholesterol levels in diet-induced obese mice. The studies also allowed to lower the level of fasting glucose, insulin, glucagon, triglycerides, and free fatty acids; in addition, there was an improvement in the level of glucose tolerance in mice with type 2 diabetes. Importantly, the administration of 11β-HSD1 inhibition slowed the progression of atherosclerotic plaques, which are the key consequences clinical conditions of the metabolic syndrome in mice with atherosclerosis. Mice deficient in apolipoprotein E and treated with an 11β-HSD1 inhibitor showed an 84% reduction in aortic cholesterol accumulation as well as lower serum cholesterol and triglyceride levels [91]. Sandeep et al. did not observe changes in glycemic control and serum lipid profile in patients aged 55–75 years with type 2 diabetes after 11β-HSD1 inhibition with carbenoxolone; however, researchers observed an improvement in verbal memory in the study group, which may be helpful in preventing the cognitive deterioration that occurs with age [65]. On this occasion, it should be mentioned that there are a number of studies that show a positive effect of the selective inhibition of 11β-HSD1 in the treatment of age-related cognitive dysfunction, which is closely related to the activity of GCs [92,93,94,95,96]. Some 11β-HSD1 inhibitors have been studied in clinical trials. Promising results have been presented in the context of the mk-0916 inhibitor in patients with type II diabetes. There was a significant decrease in glycated hemoglobin level and body weight in patients with type 2 diabetes in comparison with placebo group [68,71]. Rosenstock et al., in turn, demonstrated a positive effect of the INCB13739 inhibitor on the lipid profile of patients with type II diabetes treated with metformin [69]. The effectiveness of the inhibitors RO5093151/RO-151 and RO5027383/RO-838 was analyzed on a similar group of patients. These studies, conducted by the team of Heise et al., resulted in a reduction in body weight and the level of glycated hemoglobin, but unfortunately, RO5027383/RO-838 contributed to the deterioration of the parameters of the lipidogram (VLDL and triglycerides). In turn, the use of RO5093151/RO-151 contributed to an increase in the level of ACTH and androgen precursors [67].

Studzińska et al. conducted research on the search for selective 11β-HSD1 inhibitors among 2-aminothiazolone derivatives containing various alkyl and cyclic substituents in the 5-position of the thiazole ring and in the amino group [9,97,98,99,100].

The 2-(adamantan-1-ylamino)thiazol-4(5*H*)-one derivatives turned out to be the most active. For most of them, greater than 50% inhibition of 11β-HSD1 and less than 45% inhibition of 11β-HSD2 activity was observed at a concentration of 10 µM. It turned out that 2-(adamantan-1-ylamino)-1-thia-3-azaspiro[4.5]dec-2-en-4-one inhibits the activity of isoform 1 by 82.82%, and this value is comparable to the known carbenoxolone inhibitor. The studies were conducted using liver microsomes. The obtained results support the participation of GCs in the pathogenesis of such metabolic diseases as obesity, metabolic syndrome, or Cushing’s syndrome [101].

## 6. Summary

Research on 11β-HSD1 inhibitors proves to be a promising area for their further development and expansion, especially if the potential therapeutic benefit has not yet been fully understood. Since the level of cortisol is increased in many disease processes, the question arises regarding whether its reduction affects the underlying disease and, finally, whether it is an effect or a cause of changes in the metabolic pathway. Therefore, attempts to understand the correlation between the level of cortisol and the course of the disease seem understandable. An interesting therapeutic path is the discovery of 11β-HSD1 inhibitors that modulate cortisol levels. While the HPA axis controls the level of circulating GCs, the enzyme 11β-HSD1 acts to increase this level in cells and tissues with the participation of intrinsically neutral metabolites. Thus, the use of 11β-HSD1 inhibitors may lead to a therapeutic reduction in GC levels independent of the HPA axis. The conducted research shows that 11β-hydroxysteroid dehydrogenase type 1 inhibitors may turn out to be important substances with a therapeutic effect. The inhibition of this enzyme is a promising area for further research, especially in diseases such as diabetes, metabolic syndrome, and obesity, which very often appear in patients with age and lead to a reduction in the quality of life. Initially, these studies were conducted in animal models. Surely, their continuation in humans is a large challenge, but it allows for a better understanding of the mechanisms involved and underlying the above-mentioned diseases. It is necessary to be able to assess the inhibition of 11β-HSD1 and, consequently, the therapeutic and clinical effect in subsequent steps. Therefore, we believe that despite the stagnation observed in recent years in clinical trials of substances inhibiting the activity of 11β-HSD1, it should still be an important aspect in the search for innovative therapeutics in the treatment of metabolic diseases.

## Figures and Tables

**Figure 1 ijms-23-08984-f001:**
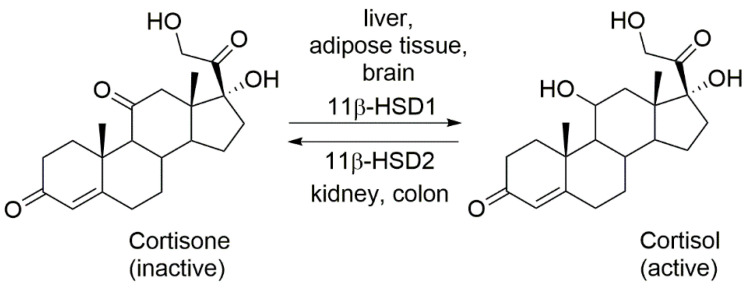
The physiological role of the two isoenzymes: 11β-HSD1 and 11β-HSD2 [9].

**Figure 2 ijms-23-08984-f002:**
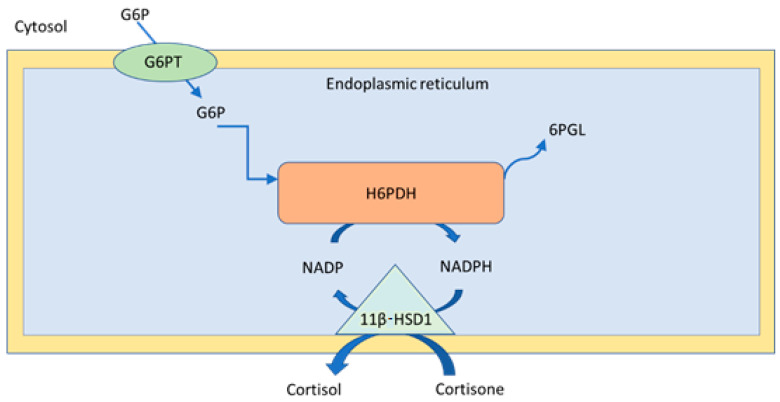
Activity of 11β-HSD1 in the presence of H6PDH.

**Figure 3 ijms-23-08984-f003:**
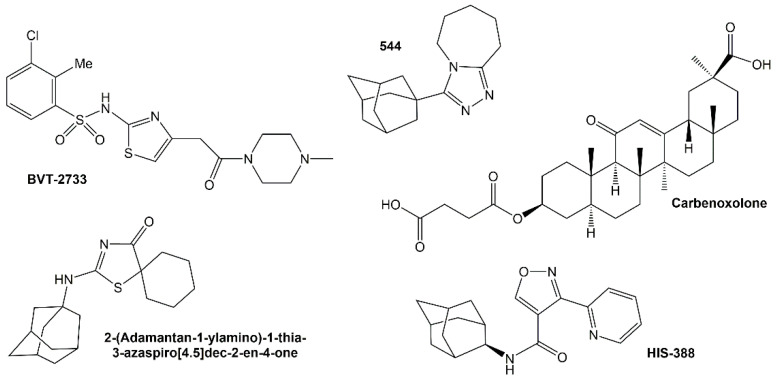
Structures of selected 11β-HSD1 inhibitors.

## Data Availability

Not applicable.

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
