# Peer review of "11β-HSD as a New Target in Pharmacotherapy of Metabolic Diseases"

_ijms, 2022, doi:10.3390/ijms23168984_

Round 1

Reviewer 1 Report

The manuscript has improved markedly, particularly by the more detailed descriptions of various clinical trials with HSD inhibitors. While I agree that a clinical development program takes a long time, I remain skeptical about the value of HSD as a valid target. If more than a decade after the first clinical data were published, no major phase III study is ongoing (every such study must be registered on clinicaltrials.gov or similar prior to start), apparently the pharmaceutical industry at large has given up on that target. While the authors do not necessarily have to agree with me or the mainstream pharmaceutical industry, I had hoped for a more balanced discussion and conclusion (the conclusion section remains virtually unchanged compared to the original submission). But it is of course the right of the authors to draw their own conclusions as long as the facts have been represented fairly; the latter is now the case.

Author Response

Dear Reviewer

We would like to thank the Reviewer for the valuable remarks and for valuable exchange of scientific ideas. We have extended the conclusion section of the manuscript, to include the information about stagnation in the field of clinical trials. English language was spellchecked by native speaker.

Reviewer 2 Report

The authors responded positively to the comments and comments.

In accordance with the comments and comments, the authors have changed their article accordingly. The manuscript has been significantly improved. I confirm that

1. The title of the article has been changed.

2. An additional keyword "metabolic syndrome" has been added to the list of keywords.

3. The purpose of the study is clearly stated.

4. In the section "Regulation of the local action of glucocorticoids", reference articles with the current level of knowledge have been added. The section is supplemented with more modern quotations. The previously cited references are described in more detail along with the details of the study.

5. The section "The role of glucocorticoids in metabolic syndrome" has been expanded. The section includes detailed biochemical mechanisms underlying the involvement of glucocorticoids in obesity and metabolic complications.

6. The section "11ß-HSD1 inhibitors in the therapy of components of the metabolic syndrome" has been supplemented with more relevant articles concerning studies of potential inhibitors.

Author Response

Dear Reviewer

We would like to thank the Reviewer for valuable remarks and acceptance of our changes in the manuscript. English language was spellchecked by native speaker.